# Sex Differences in Traditional School Bullying Perpetration and Victimization among Adolescents: A Chain-Mediating Effect

**DOI:** 10.3390/ijerph19159525

**Published:** 2022-08-03

**Authors:** Minqi Yang, Hanxiao Guo, Meimei Chu, Chongle Leng, Chunyu Qu, Kexin Tian, Yuying Jing, Mengge Xu, Xicheng Guo, Liuqi Yang, Xiaomeng Li

**Affiliations:** 1School of Education, Zhengzhou University, Zhengzhou 450001, China; minqi_yang@zzu.edu.cn (M.Y.); guohanxiao166@gmail.com (H.G.); chumeimei1538@gmail.com (M.C.); lengchongle@gmail.com (C.L.); chuny777777@gmail.com (C.Q.); xxxxxin622@gmail.com (K.T.); aspirinjyy599@gmail.com (Y.J.); a1216839506@gmail.com (M.X.); xich9423@gmail.com (X.G.); 2School of Marxism, Zhengzhou University, Zhengzhou 450001, China; 3School of Hydraulic Science and Engineering, Zhengzhou University, Zhengzhou 450001, China; liu1187612717@gmail.com

**Keywords:** sex, bullying perpetration, bullying victimization, Machiavellianism, school climate, adolescents

## Abstract

The study explored sex differences in traditional school bullying perpetration and victimization among Chinese adolescents and the effects of Machiavellianism and school climate. Data were collected from 727 adolescents (M = 16.8 years, SD = 0.9) who completed the Olweus Bully/Victim Questionnaire, Kiddie Machiavellian Scale, and School Climate Perception Questionnaire. Results showed: (1) boys were more likely to bully others and be bullied; (2) both Machiavellianism and school climate partially mediated sex differences in school bullying perpetration and victimization; (3) the chain-mediating effect of Machiavellianism and school climate on sex differences in bullying perpetration and victimization was significant. These results provide insight into the sex differences in Chinese traditional school bullying perpetration and victimization. The implications are interpreted and discussed.

## 1. Introduction

Traditional school bullying is a global public health issue with debilitating outcomes [1]. Traditional bullying, including physical contact (e.g., pushing and hitting), verbal harassment (e.g., name-calling and taunting), rumor spreading, obscene gestures, and intentionally excluding a person from a group [2], is the most common form of violence on school campuses [3]. In China, school bullying has long been a serious problem, and it has shown an escalating trend in recent years [4]. A survey showed that among 104,834 students in 22 provinces of China, the incidence of school bullying is 33.4%, with 4.7% and 28.7% of participants reported being bullied frequently or sometimes, respectively [5]. There is growing evidence that bullying and being bullied in adolescence have negative effects that persist into adulthood [6,7], with the most commonly reported effects being long-term mental health problems, such as depression, anxiety, and suicidal behavior [7,8], and psychological problems, such as poor school functioning and criminal behaviors [9].

Sex differences have been reported in the traditional school bullying perpetration and victimization. A review showed that boys are, in general, more likely to engage in bullying than girls, and boys are commonly victims and perpetrators of direct forms of bullying, while girls experience indirect bullying [10]. Some authors make a distinction between direct (overt) bullying and indirect (covert, relational) bullying. Direct bullying includes physical and verbal aggression. Direct bullying includes all sorts of physical and verbal aggression, such as kicking, hitting, threatening, name calling, and insulting. Indirect bullying includes aspects of social isolation such as ignoring, excluding, and backbiting. Direct bullying is more frequent in boys, and indirect bullying is more frequent in girls [11,12,13]. A recent study revealed the percentage of boys (45%) who reported bullying others was larger than girls (30%), and bullying victimization levels were higher for boys, mainly when considering more frequent victimization situations (five or more times), with 20% for boys and 8% for girls [14]. Although many articles have been published on the existence of sex differences, the mechanism underlying the sex differences in school bullying and victimization has not been well established. In the current study, we postulated that Machiavellianism and school climate may interpret the sex difference in bullying and victimization.

As one component of the Dark Triad, also dark personalities, which focuses on the socially aversive side of personality without being a clinical concept and is a composition of three conceptually distinct but empirically overlapping personality traits [15,16], namely Machiavellianism, narcissism (refers to feelings of pride, superiority over others, admiration-seeking, egoism, and lack of empathy), and psychopathy (used to address mostly callousness, a lack of remorse, antisocial and impulsive behaviors [16]), Machiavellianism, characterized by a deceitful, materialistic, unemotional, and selfish stance [17,18] is a behavioral tendency of individuals to use others to achieve personal goals [19]. According to the widely used Kiddie Mach scale (KMS) developed by Sutton and Keogh [20], Machiavellianism has three dimensions: distrust of humanity, dishonesty, and distrust. Studies have consistently found that men exhibit higher levels of Machiavellianism than women [21,22]. Furthermore, Machiavellianism has been reported to have an association with bullying [17,20,23,24], and bullies hold more Machiavellian attitudes [20]. Since bullying has been linked to the pursuit of dominance goals, particularly in adolescence [24], and viewed as an effective tool in handling relationships with peers [25], researchers suggested Machiavellian adolescents may perceive their relationships with peers as functional in order to achieve their goals, and thus, they may use different social behaviors (antisocial and prosocial) to manipulate their relationships [26].

According to evolutionary theories, women and men have somewhat different reproductive natures (e.g., women invest more in offspring than men do, both physiologically and behaviorally), and the two sexes evolved to have somewhat different traits, particularly in domains related to reproduction [27]. In the realm of personality, higher male levels of aggressiveness and status seeking presumably evolved as sexually selected traits that fostered male dominance and helped ancestral men attract mates, whereas higher female levels of nurturing offspring, tender-mindedness, and people orientation evolved as sexually selected traits that fostered women’s success at rearing children [28]. Additionally, according to life-history strategy (LHS) theory, which describes the trade-offs individuals make in energy allocation toward different life tasks, including bodily growth and maintenance, mating effort, and parenting/kin investment [29], if energy is allocated into survival, individuals pay attention to maintain their body, develop knowledge and skills, raise offspring, and have long-term plans, whereas if the energy is allocated into reproduction, individuals tend to show precocious puberty, have more children and less investment in raising their generations, and have a preference for immediate satisfaction and short-term benefits. The former is called “slow LHS” and the latter “fast LHS” [30]. Machiavellianism is the indicator of a fast life-history strategy [29,31]. The effect of LHS on aggressive and/or violent behaviors was exclusively indirect through evolutionary domain-specific risks, mainly mate attraction [32]. More specifically, fast LHS is related to higher risk propensity in mate attraction, which in turn increases global risk-taking behaviors, rule breaking, and dangerous, destructive, and illegal behaviors [33]. Studies have shown that men tend to have faster life-history speeds than women do and are more interested in casual sex [33], whereas women characterized by such traits may pay higher reproductive costs [34]. Based on the aforementioned empirical studies and theories, we postulated that Machiavellianism may mediate the sex differences in school bullying and victimization.

School climate is a broad, multi-dimensional concept that refers to social aspects of the learning environment, including school members’ interactions and relationships, shared values, and norms [1]. According to the social ecological theory [35], bullying is not just the result of individual characteristics such as individual biological differences and psychological traits but is also influenced by multiple relationships with peers, families, teachers, neighbors, and interactions with societal influences (e.g., media and technology) [36]. Adolescents who perceive a positive school climate are less likely to bully others [37]. Positive outcomes for adolescents are most likely to be achieved when school environments meet their developmental needs, according to the stage-environment fit theory [38]. However, when adolescents perceive a negative school climate, they are more likely to bully others [39]. Considerable research has indicated that perceived school climate is a powerful predictor of adolescents’ bullying perpetration [40].

Bronfenbrenner [35] also proposed that proximal processes, or the interactions between individuals and others within their immediate environment, have the most direct impact on development. In a school setting, these proximal processes would include students’ interactions with peers and adults within the school. Additionally, individual characteristics shape interactions between the person and other individuals and opportunities present in the environment [41]. For example, ethnicity, race, and gender influence the interactions a student has in school with other students and teachers [42,43], and these interactions may help form the student’s perception of school climate and achievement [44].

Furthermore, according to Rudasill and colleagues’ systems view of school climate (SVSC), which is situated within ecological systems theory [35] and provides a conceptual map to guide the formation of smaller causal models for individuals who experience dissonance in their home and school environments, individual characteristics and the societal expectations based on race, ethnicity, and gender are central to the experiences of students in the school environment, and individual characteristics of students are potential influences on school climate. Empirical research demonstrates that students respond differently to school climate based on individual characteristics such as race [45,46,47] and sexual orientation [48]. Given the aforementioned theories and empirical studies, we postulated that school climate would mediate the relationship between sex and bullying and victimization.

Empirical studies also explored the association between psychological traits and school climate. For example, Mitchell [49] found that perceptions of the school environment have significant relationships with personality characteristics, particularly the trait of conformity. Another study showed that except the negatively associated with neuroticism, school connectedness, one aspect of school climate, is positively associated with the other four Big Five Personality Traits (extraversion, agreeableness, conscientiousness, and openness).

Moreover, school climate could mediate the effects of personality on outcomes. For example, Irmaini and colleagues [50] using 117 senior high school principals, have demonstrated that organizational climate could mediate the effects of personality on leadership effectiveness [50]. Some researchers demonstrated psychopathy, one of the dark personalities, exerted an effect on the way students perceived aggressive attitudes and on their perceived willingness to seek help, two aspects of the school climate relevant to bullying [51], which in return affected bullying perpetration and victimization at school [52,53,54]. Another study showed that only aggressive attitudes, one aspect of school climate relevant to bullying [51], mediated the effect of psychopathy on cyberbullying [55]. Thus, we postulated that school climate may also mediate the effect of another dark personality—Machiavellianism—on bullying and victimization; that is, Machiavellianism and school climate may sequentially mediate the sex difference in bullying and victimization.

All in all, according to the literature mentioned above, the sex differences in bullying have been well reported, and the associations between personality and bullying and between school climate and bullying have also been well explored; the relationship among the four psychological constructs has seldom and the mechanisms underlying the sex differences in bullying perpetration and victimization has not well established. Thus, the aim of the current study was to explore the sex differences in the perpetration and victimization of traditional school bullying among Chinese senior high school students and the mediating effects of Machiavellianism and the school climate to further provide more constructive suggestions for prevention and intervention of school bullying. Based on the theoretical foundations and previous empirical research, we proposed three hypotheses: (1) boys may be more likely to be involved in the traditional school bullying perpetration and victimization; (2) both Machiavellianism and school climate would mediate the relationship between sex and traditional school bullying perpetration and victimization; (3) Machiavellianism and school climate may play a chain-mediating effect on the relationship between sex and bullying perpetration and victimization (Figure 1).

## 2. Materials and Methods

### 2.1. Participants and Procedure

This study employed data from a high school in Zhengzhou, China. Participants of the survey were from 20 classes, which were randomly chosen from the school. After the written informed assent was obtained from the participants and the written informed consent was obtained from the students’ guardians, the participants were assured that their answers were anonymous and confidential and were asked by teachers and well-trained psychological Ph.D. students to complete the questionnaires. After removing invalid questionnaires, the effective sample size of this panel study was 727 (M = 16.8, SD = 0.9), consisting of 371 girls (48.3%) and 356 boys (46.4%). Senior high school Grade one students accounted for 53.8%, Grade two students 35.0%, Grade three students 11.2%. In total, 69.5% of the students have their household registrations in cities, 12.0% of the students in towns, and 18.5% of the students in rural areas. The missing data of the non-demographic variables part were replaced by using the linear interpolation method. All eligible respondents were of Han Chinese race. This project was reviewed and approved by the ethics committee at the authors’ institution and had therefore been performed in accordance with the ethical standards laid down in the 1964 Declaration of Helsinki and its later amendments. The participants received payment after completing the questionnaires.

### 2.2. Measures

#### 2.2.1. Chinese Version of Kiddie Machiavellian Scale (KMS)

Pathological personality traits were measured using the brief version of KMS, which is composed of 16 items that assess distrust (five items; e.g., “*Sometimes you have to hurt others in order to get what you want*”), distrust of humanity (seven items; e.g., “*Don’t tell anyone the real reason for doing something unless you have a special purpose*”), dishonesty (four items; e.g., “*The best way to interact with someone is to say what they want to know*”). Participants answered on a 4-point Likert scale (0 = “*totally agree*”, 3 = “*totally disagree*”). Higher scores indicated lower levels of Machiavellianism. The scale has been proven to have good reliability and validity in Chinese samples [56]. In this study, Cronbach’s α coefficient is 0.71.

#### 2.2.2. School Climate Perception Questionnaire

In the current study, the school climate perception was measured by the School Climate Perception Questionnaire [57], which contains 6 items (e.g., “*There are students fighting in school*”). Participants answered on a 4-point Likert scale (1 = “*never*”, 4 = “*always*”). The total score was the sum of the scores of all items, with a higher score indicating a more negative perception of the school climate. It has been confirmed to have good reliability and validity in Chinese samples [58]. In this study, Cronbach’s α coefficient is 0.73.

#### 2.2.3. Chinese Version of the Olweus Bully/Victim Questionnaire

The Chinese version of the Olweus Bully/Victim Questionnaire mainly includes two dimensions: bullying perpetration (six items, e.g., “*Give some classmates a bad name and scold them, or make fun of them*”) and bullying victimization (six items, e.g., “*Call me a bad nickname, scold me, or make fun of me*”). Participants answered on a 5-point Likert scale (1 = “*Nothing happened in this semester*”, 5 = “*several times a week*”). Higher scores indicated a higher level of bullying perpetration or victimization. The Chinese version of the Olweus Bully/Victim Questionnaire has been recognized for its reliability and different forms of validity [59]. In this study, the Cronbach’s α coefficient for bullying perpetration is 0.79, and bullying victimization 0.76.

### 2.3. Data Analysis

Descriptive analysis was calculated to describe sociodemographic characteristics. The correlations of study variables were analyzed by Pearson correlation analyses. The mediation and chain-mediating model were analyzed using the PROCESS macro for SPSS [60]; 95% confidence intervals that do not contain zero indicate significant mediation effects [60]. To determine whether an indirect effect was statistically significant, we used maximum likelihood estimation and the bias-corrected bootstrap 95% confidence interval based on 5000 bootstrapping. Model 6 was used to examine the chain-mediating effect.

## 3. Results

### 3.1. Descriptive Results

Means and standard deviations among the primary variables in the study are shown in Table 1. Boys were found to score significantly lower on Machiavellianism (*t* = −3.92, *p* < 0.001), which indicated that boys have higher Machiavellianism levels than girls. Boys were found to score significantly higher on school climate (*t* = 4.03, *p* < 0.001), bullying perpetration (*t* = 2.58, *p* < 0.05), and bullying victimization (*t* = 4.78, *p* < 0.001), which indicated that boys have a more negative perception of school climate, higher level of bullying perpetration and victimization than girls.

### 3.2. Correlation Results

As shown in Table 2, Machiavellianism was significantly negatively correlated with school climate, traditional school bullying perpetration and victimization (*r* = −0.22, *p* < 0.001, *r* = −0.21, *p* < 0.001; *r* = −0.17, *p* < 0.001). However, school climate was significantly positively correlated with traditional school bullying perpetration and victimization (*r* = 0.25, *p* < 0.001; *r* = 0.34, *p* < 0.001), and bullying was significantly positively correlated with victimization (*r* = 0.53, *p* < 0.001).

### 3.3. Chain-Mediating Effect

To avoid multicollinearity, all variables were standardized prior to the analysis. Sex was dummy coded such that 0 = male and 1 = female. The chain-mediation analyses were conducted using PROCESS for SPSS (version 25.0, IBM Corporation, Armonk, NY, USA) (model 6) [60].

In terms of bullying perpetration, the result showed that Machiavellianism significantly negatively predicted bullying perpetration (*β* = −0.16, *t* = −4.38, *p* < 0.001), school climate significantly positively predicted bullying perpetration (*β* = 0.22, *t* = 5.88, *p* < 0.001), but the direct effect of sex on bullying perpetration was not significant. The mediating effect was −0.113, with 95% CI [−0.214, −0.054] not including zero, which indicated that Machiavellianism played a mediating role in the association between sex and school climate, school climate also played a mediating role between the association between Machiavellianism and bullying victimization, and Machiavellianism and school climate chain-mediated the sex difference in the bullying victimization (Figure 2).

Figure 3 showed sex and Machiavellianism significantly negatively predicted bullying victimization (*β* = −0.24, *t* = 3.35, *p* < 0.001; *β* = −0.10, *t* = −2.72, *p* < 0.05), and school climate significantly positively predicted bullying victimization (*β* = 0.30, *t* = 4.48, *p* < 0.001). The total mediating effect was −0.119, while the 95% CI was [−0.208, −0.062] not including zero, which indicated that Machiavellianism played a partial mediating role between the association between sex and school climate, school climate also played a partial mediating role between the association between Machiavellianism and bullying victimization, and Machiavellianism and school climate partially chain-mediated the sex difference in the bullying victimization.

## 4. Discussion

The present study explored differences in traditional school bullying perpetration and victimization behaviors between boys and girls and the effects of Machiavellianism and school climate. Consistent with our hypothesis, boys were more likely to be involved in the traditional school bullying perpetration and victimization, and sex indirectly influenced traditional school bullying perpetration and victimization behaviors via Machiavellianism and school climate. Furthermore, a novel chain-mediation model revealed that Machiavellianism and school climate mediated the effects of sex on traditional school bullying perpetration and victimization behaviors.

The present study revealed the sex differences in traditional school bullying perpetration and victimization, and the sex had a direct effect on bullying victimization, which is consistent with a previous study showing the level of traditional school bullying perpetration and victimization by boys were generally higher than girls [61]. Some researchers posted that the aggressive tendency of males is the result of the interaction between biological and physiological factors, social and psychological factors, and specific situations [62] and is also related to normative expectations and the anticipated consequences of aggression, that is, if boys display aggressive behaviors, they may suffer social consequences [63,64]. Additionally, the evolutionary approach to bullying behavior treats bullying, like other common forms of aggressive behavior, as having costs and benefits and as being, in some circumstances, adaptive for an individual doing the bullying (e.g., by gaining resources or defending sub-group identity), even if not beneficial for the victim or the wider community. The sex differences in bullying and aggression can be explained in terms of sexual selection theory and effective strategies for damaging the status or reputation of males or females, respectively [65].

The present study showed that, as had been hypothesized, Machiavellianism mediated the sex differences in bullying. Specifically, boys are prone to develop a higher level of Machiavellianism with which one is more likely to possess self-focused goals and aggressive strategies [66], and they are more likely to be involved in aggressive behaviors. This result also supported the life-history-strategy (LHS) theory, individuals higher in Machiavellianism as the indicator of the “fast LHS” [29,67], which is reflective of reproductive efforts over somatic efforts [68] tend to conduct aggressive and/or violent behaviors, as a measure to accomplish evolutionary domain-specific tasks, mainly mate attraction [32]. Researchers found that fast LHS is related to higher risk propensity in mate attraction, which in turn increases rule breaking and dangerous, destructive, and illegal behaviors [32].

This study also reveals that, as hypothesized, the perception of school climate mediates sex differences in school bullying, which indicates that having a worse perception of school climate may also be the reason for the sex differences in bullying. The result was consistent with previous studies. For example, one study showed promoting a positive school climate could reduce bullying perpetration [69]. Some researchers have found that a positive school climate protects adolescents against bullying victimization [70,71]. There is also research indicating that when students perceived a more positive school climate, the negative association between bullying victimization and student engagement was stronger [3]. Students who attend schools with positive school climates are more likely to seek support upon experiencing victimization and threats of violence compared to students with negative school climates [72]. In addition, some longitudinal studies have revealed that school climate, including student–teacher relationships, clear expectations, and fairness of rules, significantly predicts bullying victimization [71,73]. This result could be explained by the social ecological theory [35], which posts that bullying is not just the result of individual characteristics but is influenced by multiple relationships with peers, families, and teachers, that is, school climate which has the most direct impact on development. Additionally, individual biological differences, such as sex and race, shape interactions between the person and other individuals and opportunities present in the environment [47], which influences bullying perpetration and victimization [35].

More importantly, the current study revealed, as has been hypothesized, that Machiavellianism and school climate played a chain-mediating effect on the sex differences in bullying perpetration and victimization, which indicated that boys tend to display a higher level of Machiavellianism, and then they are more likely to have a negative perception of school climate and ultimately be involved in the bullying perpetration and victimization. The results were consistent with previous studies; for example, researchers have demonstrated psychopathy, which shared some common features with Machiavellianism, exerted an effect on the school climate relevant to bullying [51], which further affected bullying perpetration and victimization at school [52,53,54]. The result supports the general aggression model posting individuals’ personality factors may affect their cognition and then affect their aggressive behaviors [74]. Past studies have shown that students are less likely to be engaged in aggression and bullying when they perceive their friends as trustworthy and helpful and their school climate as trusting and pleasant [37,75]. This result could also be explained by the social ecological theory [35], which posts that p shape interactions between the person and other individuals and opportunities present in the environment [47], which influences bullying perpetration and victimization [35].

## 5. Limitations

Despite the novelty and theoretical rigor of our study, several limitations inherent to the present study are worth mentioning. Firstly, the use of self-reported data may cause potential bias. Future research may use additional data sources, such as reports from family members, teachers, and friends. Secondly, we only used the sample from senior high school students. Future research should explore the generalizability of this model in more diverse samples, such as samples from primary and middle school students, which may provide greater variation in traditional school bullying perpetration and victimization. Thirdly, the present study explored the mediating roles of Machiavellianism, one of the components of the Dark Triad; the other two components of the Dark Triad, that is, psychopathy and narcissism, could also be explored in future studies.

## 6. Implications

The present results revealed the mechanism underlying the sex differences in traditional school bullying perpetration and victimization among Chinese adolescents, and the findings have important implications for intervention for school bullying. Firstly, boys are more likely to be involved in the traditional school bullying perpetration and victimization; the educators and parents could pay more attention to the schoolboys rather than girls when trying to reduce the bullying phenomenon in the school. For example, parents and educators could conduct more bullying education than girls, give more encouragement to boys to deal with relationships with peers nonviolently and amicably and to further perceive the school climate more positively, and try to conduct more bullying prevention and intervention programs among boys. Secondly, Machiavellianism and school climate may be the reasons for the sex differences in bullying perpetration and victimization, which indicates that intervention for reducing the level of Machiavellianism and enhancing the positive level of school climate (for example, all school members are enlisted in an effort to combat bullying) may help reducing traditional school bullying perpetration and victimization among the adolescents. Thirdly, the current results showed that Machiavellianism and school climate played a chain-mediating role in the relationship between sex and bullying, which indicated that sex, Machiavellianism, and school climate jointly affect the traditional school bullying perpetration and victimization among Chinese adolescents. Therefore, all these aspects should be taken into account in future interventions to achieve the best intervention effect and finally to reduce the level of students’ bullying perpetration and victimization.

## 7. Conclusions

This study revealed mediating and chain-mediating effects of Machiavellianism and school climate on the sex differences in traditional school bullying perpetration and victimization among Chinese adolescents.

## Figures and Tables

**Figure 1 ijerph-19-09525-f001:**
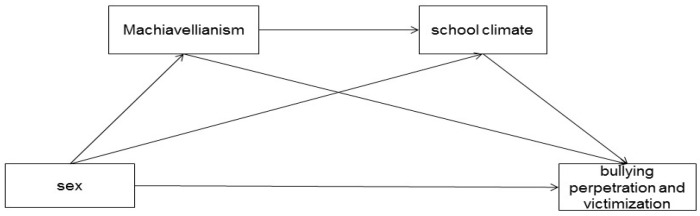
Hypothesized relationships among sex, Machiavellianism, school climate, and bullying perpetration and victimization.

**Figure 2 ijerph-19-09525-f002:**
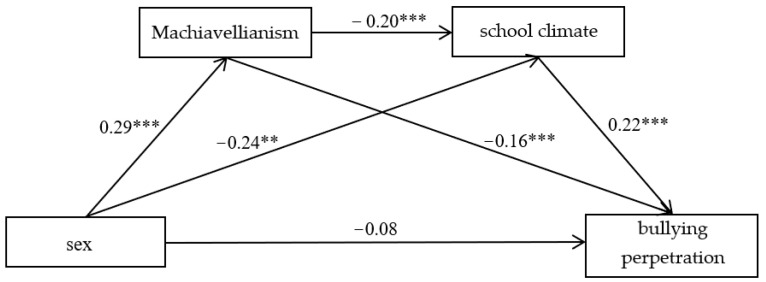
The chain-mediating effect of Machiavellianism and school climate between sex and bullying perpetration. Note: ** *p* < 0.01, *** *p* < 0.001.

**Figure 3 ijerph-19-09525-f003:**
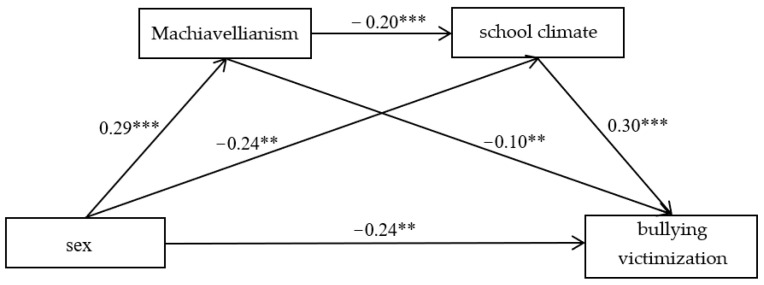
The chain-mediating effect of Machiavellianism and school climate between sex and bullying victimization. Note: ** *p* < 0.01, *** *p* < 0.001.

**Table 1 ijerph-19-09525-t001:** Means, standard deviations, and sex differences among the primary variables.

	Total	Boys	Girls	*t*	*p*
Variables	(*n* = 727)	(*n* = 356)	(*n* = 371)		
	M ± SD	M ± SD	M ± SD		
1. Machiavellianism	43.03 ± 6.00	42.20 ± 6.17	43.93 ± 5.73	−3.92	<0.001
2. School climate	9.01 ± 2.97	9.47 ± 3.39	8.58 ± 2.49	4.03	<0.001
3. Bullying perpetration	6.64 ± 2.26	6.88 ± 2.68	6.44 ± 1.86	2.58	<0.05
4. Bullying victimization	7.37 ± 2.82	7.90 ± 3.44	6.89 ± 2.05	4.78	<0.001

**Table 2 ijerph-19-09525-t002:** Pearson correlation coefficients between main variables (*n* = 727).

Variables	1	2	3	4
1. Machiavellianism	1			
2. School climate	−0.21 ***	1		
3. Bullying perpetration	−0.21 ***	0.25 ***	1	
4. Bullying victimization	−0.17 ***	0.34 ***	0.53 ***	1

Note: *** *p* < 0.001.

## Data Availability

The datasets generated during and/or analyzed during the current study are available from the corresponding author on reasonable request.

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
