# Peer review of "Sex Differences in Traditional School Bullying Perpetration and Victimization among Adolescents: A Chain-Mediating Effect"

_ijerph, 2022, doi:10.3390/ijerph19159525_

Round 1

Reviewer 1 Report

Introduction

The study explores an interesting subject taking into account the mediating effects proposed. However, the study's gap is unclear, and the study's novelty is not well established. Also, the mediating effects are not clearly stated or explained.

Sex differences have been widely explored in literature, also the role of school climate in bullying and victimization. 

It is important to explain o give examples of direct and indirect bullying. The justification for the differences attributed to the reproductive nature is weak. More citations are needed to support these claims, or a substantial body of studies must be added.

The mediating role of Machiavellianism and school climate is not well-established. 

Machiavellianism and the school climate chain mediating effect also is not well-established. 

It is important to clarify the aim of the study. 

Method.

More information about the sample is necessary, such as socio-economic information. 

Sampling information is missing. 

Dimensions of Machiavelism are in the method section but not defined in the introduction. 

Evidence about the validity of the scales is required. 

Results

Also is important to check this affirmation". Still, the direct effect of sex on bullying perpetration was not significant, which indicated that Machiavellianism played a mediating role in the association between sex and school climate" the relationship could be significant and play a mediating role. 

Discussion

In the results section, the authors indicate that the direct effect of sex on bullying perpetration was not significant. However, in the discussion, the authors affirm that "sex not only directly influences traditional school bullying perpetration and victimization behaviors but also indirectly via Machiavellianism and school climate." How is this possible? 

The authors should align the discussion to the results presented.

The authors must also explain the positive association between school climate and bullying and victimization in the discussion.

Author Response

Subject:

Sex differences in traditional school bullying perpetration and victimization among adolescents: A chain-mediating effect

Response letter

Dear editor,

Thank you for giving us the opportunity to revise our manuscript. We appreciate the reviewers for the helpful and constructive comments and suggestions, which are of great value to improving the quality of our article. Based on the comments, careful modifications have been made to our previous draft. In the current draft, all changes were highlighted within the document by blue colored text. After this revision, we also wrote a point-by-point response letter to the three reviewers’ comments.

We hope the revised manuscript will be acceptable for publication in the International Journal of Environmental Research and Public Health. Thank you very much for your time.

Reviewer 1

Open Review

(x) I would not like to sign my review report

( ) I would like to sign my review report

English language and style

( ) Extensive editing of English language and style required

(x) Moderate English changes required

( ) English language and style are fine/minor spell check required

( ) I don't feel qualified to judge about the English language and style

Yes

Can be improved

Must be improved

Not applicable

Does the introduction provide sufficient background and include all relevant references?

( )

( )

(x)

( )

Are all the cited references relevant to the research?

( )

(x)

( )

( )

Is the research design appropriate?

(x)

( )

( )

( )

Are the methods adequately described?

( )

(x)

( )

( )

Are the results clearly presented?

(x)

( )

( )

( )

Are the conclusions supported by the results?

( )

(x)

( )

( )

Comments and Suggestions for Authors

Introduction

  1. The study explores an interesting subject taking into account the mediating effects proposed. However, the study's gap is unclear, and the study's novelty is not well established. Also, the mediating effects are not clearly stated or explained.

Response: Thanks for your suggestions and we accept the suggestion. We have rewritten the introduction about the mediating effects, please check the revised sentences in blue color are like this:

  • the study's gap is unclear, and the study's novelty

“All in all, according the literature mentioned above, the sex differences in bullying has been well reported, and the associations between personality and bullying and between school climate and bullying have been also well explored, the relationship among the four psychological constructs has seldom and the mechanisms underlying the sex differences in bullying perpetration and victimization has not well established. Thus, the aim of the current study was to explore the sex differences in the perpetration and victimization of traditional school bullying among Chinese senior high school students and the mediating effects of Machiavellianism and the school climate, to further provide more constructive suggestions for prevention and intervention of school bullying.”

  • the mediating effects

“Sex differences have been reported in the traditional school bullying perpetration and victimization. A review showed that boys are in general more likely to engage in bullying than girls, and boys are commonly victims and perpetrators of direct forms of bullying, while girls experience indirect bullying (Hong & Espelage, 2012). Some authors make a distinction between direct (overt) bullying and indirect (covert, relational) bullying. Direct bullying includes physical and verbal aggression. Direct bullying includes all sorts of physical and verbal aggression, such as kicking, hitting, threatening, name-calling, and insulting. Indirect bullying includes aspects of social isolation such as ignoring, excluding, and backbiting. Direct bullying is more frequent in boys, indirect bullying more frequent in girls (Björkqvist, K., Lagerspetz, K. M., & Kaukiainen, A., 1992; Rivers, I., & Smith, P. K.,1994; Van der Wal, M. F., De Wit, C. A., & Hirasing, R. A.,2003). A recent study revealed the percentage of boys (45%) who reported bullying others was larger than girls (30%), and bullying victimization levels were higher for boys, mainly when considering more frequent victimization situations (five or more times), with 20% for boys and 8% for girls (Iossi Silva, Pereira, Mendonça, Nunes, & Oliveira, 2013). Although many articles have been published on the existence of sex differences, the mechanism underlying the sex differences in school bullying and victimization has not been well established. In the current study, we postulated that Machiavellianism and school climate may interpret the sex difference in bullying and victimization.

As one component of the Dark Triad, also dark personalities, which focuses on the socially aversive side of personality without being a clinical concept and is a composition of three conceptually distinct but empirically overlapping personality traits (Furnham, Richards, & Paulhus, 2013; Rahafar, Randler, Castellana, & Kausch, 2017), namely Machiavellianism, narcissism (refers to feelings of pride, superiority over others, admiration-seeking, egoism, and lack of empathy), and psychopathy (used to address mostly callousness, a lack of remorse, antisocial and impulsive behaviors (Rahafar et al., 2017)), Machiavellianism, characterized by a deceitful, materialistic, unemotional, and selfish stance (Christie, Geis, Festinger, & Schachter, 1970; Láng & Lénárd, 2015) is a behavioral tendency of individuals to use others to achieve personal goals (Zhao & Liao, 2013). According to the widely used Kiddie Mach scale (KMS) developed by Sutton and Keogh (2001), Machiavellianism has three dimensions: distrust of humanity, dishonest, and distrust. Studies have consistently found that men exhibit higher levels of Machiavellianism than women (Paulhus & Williams, 2002; Yang et al., 2019). Furthermore, Machiavellianism has been reported to have association with bullying (Christie & Geis, 1970; Sutton & Keogh, 2001; Berger, C., & Caravita, S. C., 2016), and bullies hold more Machiavellian attitudes (Sutton & Keogh, 2000). Since bullying has been linked to the pursuit of dominance goals particularly in adolescence (Caravita & Cillessen, 2012), and viewed as an effective tool in handling relationships with peers (Garandeau & Cillessen, 2006), researchers suggested Machiavellian adolescents may perceive their relationships with peers as functional in order to achieve their goals, and thus they may use different social behaviors (antisocial and prosocial) to manipulate their relationships (Bereczkei, Birkas, & Kerekes, 2015).

According to evolutionary theories, women and men have somewhat different reproductive natures (e.g., women invest more in offspring than men do, both physiologically and behaviorally), the two sexes evolved to have somewhat different traits, particularly in domains related to reproduction (Buss, 2008). In the realm of personality, higher male levels of aggressiveness and status-seeking presumably evolved as sexually selected traits that fostered male dominance and helped ancestral men attract mates, whereas higher female levels of nurturing offspring, tender-mindedness, and people orientation evolved as sexually selected traits that fostered women’s success at rearing children (Lippa, 2010).Besides, according to life history strategy (LHS) theory, which describes the trade-offs individuals make in energy allocation toward different life tasks including bodily growth and maintenance, mating effort, and parenting/kin investment (Mcdonald, Donnellan, & Navarrete, 2012), if energy is allocated into survival, individuals will pay attention to maintain their body, develop knowledge and skills, raise offspring, and have long-term plans; whereas if the energy is allocated into reproduction, individual tend to show precocious puberty, have more children and less investment in raising their generations, and has preference for immediate satisfaction and short-term benefits. The former is called "slow LHS", the latter "fast LHS" (Jonason et al., 2015). Machiavellianism is the indicator of a fast life history strategy (Jonason & Tost, 2010; McDonald et al., 2012). The effect of LHS on aggressive and/or violent behaviors was exclusively indirect through evolutionary domain-specific risks, mainly mate attraction (Salas-Rodríguez, Gómez-Jacinto, & Hombrados-Mendieta, 2021). More specifically, fast LHS is related to higher risk propensity in mate attraction, which in turn increases global risk-taking behaviors, rule breaking, dangerous, destructive and illegal behaviors (Salas-Rodríguez et al., 2021). Studies have shown that men tend to have faster life history speeds than women do, and are more interested in casual sex (Al., 2017; Jonason & Fisher, 2009) (Al., 2017; Jonason & Fisher, 2009), whereas women characterized by such traits may pay higher reproductive costs (Jonason & Lavertu, 2017). Based on the aforementioned empirical studies and theories, we postulated that Machiavellianism may mediate the sex differences in the school bullying and victimization.

School climate is a broad, multi-dimensional concept that refers to social aspects of the learning environment including school members’ interactions and relationships, shared values and norms (Cohen et al., 2009). According to the social ecological theory (Bronfenbrenner, 1979), bullying is not just the result of individual characteristics such as individual biological differences, and psychological traits, but is influenced by multiple relationships with peers, families, teachers, neighbors, and interactions with societal influences (e.g., media, technology) (Swearer & Hymel, 2015). Adolescents who perceive a positive school climate were less likely to bully others (Yang, Wang, & Lei, 2019). Positive outcomes for adolescents are most likely to be achieved when school environments meet their developmental needs, according to the stage-environment fit theory (Gutman & Eccles, 2007). However, when adolescents perceive a negative school climate, they are more likely to bully others (Espelage, Low, et al., 2014).Considerable research has indicated that perceived school climate is a powerful predictor of adolescents’ bullying perpetration (Chan & Wong, 2015).

Bronfenbrenner (1979) also proposed that proximal processes, or the interactions between individuals and others within their immediate environment, have the most direct impact on development. In a school setting, these proximal processes would include students' interactions with peers and adults within the school. Besides, individual characteristics shape interactions between the person and other individuals and opportunities present in the environment (Bronfenbrenner, 1989). For example, ethnicity, race, and gender influence the interactions a student has in school with other students and teachers (Mason et al. 1994; (Parsons, 2008), and these interactions may help form the student’s perception of school climate and achievement (Rudasill et al., 2018).

Furthermore, according to Rudasill and colleagues’ systems view of school climate (SVSC), which is situated within ecological systems theory (Bronfenbrenner, 1979) and provides a conceptual map to guide the formation of smaller causal models for individuals who experience dissonance in their home and school environments, individual characteristics and the societal expectations based on race, ethnicity, and gender are central to the experiences of students in the school environment, and individual characteristics of students are potential influences on school climate. Empirical research demonstrates that students respond differently to school climate based on individual characteristics such as race (Bottiani et al., 2014; Spencer, 1999; Thapa et al., 2013) and sexual orientation (Fedewa & Ahn 2011). Given the aforementioned theories and empirical studies, we postulated that school climate would mediate the relationship between sex and bullying and victimization.

Empirical studies also explored the association between psychological traits and school climate. For example, Mitchell (1968) found that perceptions of the school environment have significant relationships with personality characteristics, particularly the trait of conformity. Another study showed that except the negatively associated with Neuroticism, school connectedness, one aspect of school climate, is positively associated with the other four Big Five Personality Traits (Extraversion, Agreeableness, Conscientiousness, and Openness).

What’s more, school climate could mediated the effects of personality on outcomes. For example, Irmaini and colleagues (2017) using 117 senior high school principals have demonstrated that organization climate could mediate the effects of personality on the leadership effectiveness (Irmaini, et al., 2017). Some researchers demonstrated psychopathy, one of the dark personalities, exerted an effect on the way students perceived aggressive attitudes and on their perceived willingness to seek help, two aspects of the school climate relevant to bullying (Bandyopadhyay et al., 2009), which in return affected bullying perpetration and victimization at school (Bayar & Ucanok, 2012; Gini, 2008; Kokkinos et al., 2010). Another study showed that the only aggressive attitudes, one aspect of school climate relevant to bullying (Bandyopadhyay et al., 2009), mediated the effect of psychopathy on cyberbullying (Charalampous et al., 2021). Thus we postulated that school climate may also mediate the effect of another dark personality --Machiavellianism -- on bullying and victimization, that is, Machiavellianism and school climate may sequentially mediate the sex difference in bullying and victimization.”

  1. Sex differences have been widely explored in literature, also the role of school climate in bullying and victimization. It is important to explain and give examples of direct and indirect bullying.

Response: Thanks for your suggestions and we accept the suggestion, so we added the following information to explain and give examples of direct and indirect bullying in blue color are like this:

“Some authors make a distinction between direct (overt) bullying and indirect (covert, relational) bullying. Direct bullying includes physical and verbal aggression. Direct bullying includes all sorts of physical and verbal aggression, such as kicking, hitting, threatening, name-calling, and insulting. Indirect bullying includes aspects of social isolation such as ignoring, excluding, and backbiting. Direct bullying is more frequent in boys, indirect bullying more frequent in girls (Björkqvist, K., Lagerspetz, K. M., & Kaukiainen, A., 1992; Rivers, I., & Smith, P. K.,1994; Van der Wal, M. F., De Wit, C. A., & Hirasing, R. A.,2003).”

  1. The justification for the differences attributed to the reproductive nature is weak. More citations are needed to support these claims, or a substantial body of studies must be added.

Response: Thanks for your suggestions, we agree that the differences attributed to the reproductive nature is weak, but the aim of the study was to explore the mediating effects of Machiavellianism and school climate on the sex differences in bullying, rather than the direct effect of sex on bullying, after careful consideration, we decided to delete the justification for the sex difference in the Introduction, but moved the information to the Discussion to interpret the direct effect of sex on bullying victimization, and we also added some information to interpret the sex difference, please check the following in blue color:

“The present study revealed the sex differences in traditional school bullying perpetration and victimization, and the sex had direct effect on bullying victimization, which is consistent with a previous study showing the level of traditional school bullying perpetration and victimization by boys were generally higher than girls (Pontes, Ayres, Lewandowski, & Pontes, 2018). Some researchers posted that aggressive tendency of male is the result of the interaction between biological and physiological factors, social and psychological factors and specific situations (Zhang, Gu, Wang, Wang, & Jones, 2000), and is also related to normative expectations and the anticipated consequences of aggression, that is, if boys display aggressive behaviors, they may suffer social consequences (Card, Stucky, Sawalani & Little, 2008). Besides, the evolutionary approach to bullying behavior treats bullying, like other common forms of aggressive behavior, as having costs and benefits and as being in some circumstances adaptive for an individual doing the bullying (e.g., by gaining resources or defending sub-group identity), even if not beneficial for the victim or the wider community. And the sex differences in bullying and aggression can be explained in terms of sexual selection theory, and effective strategies for damaging the status or reputation of males or females respectively (Pellegrini & Archer, 2005).”

  1. The mediating role of Machiavellianism and school climate is not well-established. 

Response: Thanks for your suggestions and we accept the suggestion, so we added some theories and empirical studies to rationalize the mediating effects of Machiavellianism and the school climate, please check them in the following information in blue color:

 “Sex differences have been reported in the traditional school bullying perpetration and victimization. A review showed that boys are in general more likely to engage in bullying than girls, and boys are commonly victims and perpetrators of direct forms of bullying, while girls experience indirect bullying (Hong & Espelage, 2012). Some authors make a distinction between direct (overt) bullying and indirect (covert, relational) bullying. Direct bullying includes physical and verbal aggression. Direct bullying includes all sorts of physical and verbal aggression, such as kicking, hitting, threatening, name-calling, and insulting. Indirect bullying includes aspects of social isolation such as ignoring, excluding, and backbiting. Direct bullying is more frequent in boys, indirect bullying more frequent in girls (Björkqvist, K., Lagerspetz, K. M., & Kaukiainen, A., 1992; Rivers, I., & Smith, P. K.,1994; Van der Wal, M. F., De Wit, C. A., & Hirasing, R. A.,2003). A recent study revealed the percentage of boys (45%) who reported bullying others was larger than girls (30%), and bullying victimization levels were higher for boys, mainly when considering more frequent victimization situations (five or more times), with 20% for boys and 8% for girls (Iossi Silva, Pereira, Mendonça, Nunes, & Oliveira, 2013). Although many articles have been published on the existence of sex differences, the mechanism underlying the sex differences in school bullying and victimization has not been well established. In the current study, we postulated that Machiavellianism and school climate may interpret the sex difference in bullying and victimization.

As one component of the Dark Triad, also dark personalities, which focuses on the socially aversive side of personality without being a clinical concept and is a composition of three conceptually distinct but empirically overlapping personality traits (Furnham, Richards, & Paulhus, 2013; Rahafar, Randler, Castellana, & Kausch, 2017), namely Machiavellianism, narcissism (refers to feelings of pride, superiority over others, admiration-seeking, egoism, and lack of empathy), and psychopathy (used to address mostly callousness, a lack of remorse, antisocial and impulsive behaviors (Rahafar et al., 2017)), Machiavellianism, characterized by a deceitful, materialistic, unemotional, and selfish stance (Christie, Geis, Festinger, & Schachter, 1970; Láng & Lénárd, 2015) is a behavioral tendency of individuals to use others to achieve personal goals (Zhao & Liao, 2013). According to the widely used Kiddie Mach scale (KMS) developed by Sutton and Keogh (2001), Machiavellianism has three dimensions: distrust of humanity, dishonest, and distrust. Studies have consistently found that men exhibit higher levels of Machiavellianism than women (Paulhus & Williams, 2002; Yang et al., 2019). Furthermore, Machiavellianism has been reported to have association with bullying (Christie & Geis, 1970; Sutton & Keogh, 2001; Berger, C., & Caravita, S. C., 2016), and bullies hold more Machiavellian attitudes (Sutton & Keogh, 2000). Since bullying has been linked to the pursuit of dominance goals particularly in adolescence (Caravita & Cillessen, 2012), and viewed as an effective tool in handling relationships with peers (Garandeau & Cillessen, 2006), researchers suggested Machiavellian adolescents may perceive their relationships with peers as functional in order to achieve their goals, and thus they may use different social behaviors (antisocial and prosocial) to manipulate their relationships (Bereczkei, Birkas, & Kerekes, 2015).

According to evolutionary theories, women and men have somewhat different reproductive natures (e.g., women invest more in offspring than men do, both physiologically and behaviorally), the two sexes evolved to have somewhat different traits, particularly in domains related to reproduction (Buss, 2008). In the realm of personality, higher male levels of aggressiveness and status-seeking presumably evolved as sexually selected traits that fostered male dominance and helped ancestral men attract mates, whereas higher female levels of nurturing offspring, tender-mindedness, and people orientation evolved as sexually selected traits that fostered women’s success at rearing children (Lippa, 2010).Besides, according to life history strategy (LHS) theory, which describes the trade-offs individuals make in energy allocation toward different life tasks including bodily growth and maintenance, mating effort, and parenting/kin investment (Mcdonald, Donnellan, & Navarrete, 2012), if energy is allocated into survival, individuals will pay attention to maintain their body, develop knowledge and skills, raise offspring, and have long-term plans; whereas if the energy is allocated into reproduction, individual tend to show precocious puberty, have more children and less investment in raising their generations, and has preference for immediate satisfaction and short-term benefits. The former is called "slow LHS", the latter "fast LHS" (Jonason et al., 2015). Machiavellianism is the indicator of a fast life history strategy (Jonason & Tost, 2010; McDonald et al., 2012). The effect of LHS on aggressive and/or violent behaviors was exclusively indirect through evolutionary domain-specific risks, mainly mate attraction (Salas-Rodríguez, Gómez-Jacinto, & Hombrados-Mendieta, 2021). More specifically, fast LHS is related to higher risk propensity in mate attraction, which in turn increases global risk-taking behaviors, rule breaking, dangerous, destructive and illegal behaviors (Salas-Rodríguez et al., 2021). Studies have shown that men tend to have faster life history speeds than women do, and are more interested in casual sex (Al., 2017; Jonason & Fisher, 2009) (Al., 2017; Jonason & Fisher, 2009), whereas women characterized by such traits may pay higher reproductive costs (Jonason & Lavertu, 2017). Based on the aforementioned empirical studies and theories, we postulated that Machiavellianism may mediate the sex differences in the school bullying and victimization.

School climate is a broad, multi-dimensional concept that refers to social aspects of the learning environment including school members’ interactions and relationships, shared values and norms (Cohen et al., 2009). According to the social ecological theory (Bronfenbrenner, 1979), bullying is not just the result of individual characteristics such as individual biological differences, and psychological traits, but is influenced by multiple relationships with peers, families, teachers, neighbors, and interactions with societal influences (e.g., media, technology) (Swearer & Hymel, 2015). Adolescents who perceive a positive school climate were less likely to bully others (Yang, Wang, & Lei, 2019). Positive outcomes for adolescents are most likely to be achieved when school environments meet their developmental needs, according to the stage-environment fit theory (Gutman & Eccles, 2007). However, when adolescents perceive a negative school climate, they are more likely to bully others (Espelage, Low, et al., 2014).Considerable research has indicated that perceived school climate is a powerful predictor of adolescents’ bullying perpetration (Chan & Wong, 2015).

Bronfenbrenner (1979) also proposed that proximal processes, or the interactions between individuals and others within their immediate environment, have the most direct impact on development. In a school setting, these proximal processes would include students' interactions with peers and adults within the school. Besides, individual characteristics shape interactions between the person and other individuals and opportunities present in the environment (Bronfenbrenner, 1989). For example, ethnicity, race, and gender influence the interactions a student has in school with other students and teachers (Mason et al. 1994; (Parsons, 2008), and these interactions may help form the student’s perception of school climate and achievement (Rudasill et al., 2018).

Furthermore, according to Rudasill and colleagues’ systems view of school climate (SVSC), which is situated within ecological systems theory (Bronfenbrenner, 1979) and provides a conceptual map to guide the formation of smaller causal models for individuals who experience dissonance in their home and school environments, individual characteristics and the societal expectations based on race, ethnicity, and gender are central to the experiences of students in the school environment, and individual characteristics of students are potential influences on school climate. Empirical research demonstrates that students respond differently to school climate based on individual characteristics such as race (Bottiani et al., 2014; Spencer, 1999; Thapa et al., 2013) and sexual orientation (Fedewa & Ahn 2011). Given the aforementioned theories and empirical studies, we postulated that school climate would mediate the relationship between sex and bullying and victimization.”

  1. Machiavellianism and the school climate chain mediating effect also is not well-established. 

Response: Thanks for your suggestions and we accept the suggestion, so we added some theories and empirical studies to rationalize the chain mediating effects of Machiavellianism and the school climate, please check them in the following information in blue color:

“Empirical studies also explored the association between psychological traits and school climate. For example, Mitchell (1968) found that perceptions of the school environment have significant relationships with personality characteristics, particularly the trait of conformity. Another study showed that except the negatively associated with Neuroticism, school connectedness, one aspect of school climate, is positively associated with the other four Big Five Personality Traits (Extraversion, Agreeableness, Conscientiousness, and Openness).

What’s more, school climate could mediated the effects of personality on outcomes. For example, Irmaini and colleagues (2017) using 117 senior high school principals have demonstrated that organization climate could mediate the effects of personality on the leadership effectiveness (Irmaini, et al., 2017). Some researchers demonstrated psychopathy, one of the dark personalities, exerted an effect on the way students perceived aggressive attitudes and on their perceived willingness to seek help, two aspects of the school climate relevant to bullying (Bandyopadhyay et al., 2009), which in return affected bullying perpetration and victimization at school (Bayar & Ucanok, 2012; Gini, 2008; Kokkinos et al., 2010). Another study showed that the only aggressive attitudes, one aspect of school climate relevant to bullying (Bandyopadhyay et al., 2009), mediated the effect of psychopathy on cyberbullying (Charalampous et al., 2021). Thus we postulated that school climate may also mediate the effect of another dark personality --Machiavellianism -- on bullying and victimization, that is, Machiavellianism and school climate may sequentially mediate the sex difference in bullying and victimization.”

  1. It is important to clarify the aim of the study. 

Response: Thanks for your suggestions and we accept the suggestion. Please see the revised sentences (p4. Line 158-161) in blue color are like this:

“All in all, according the literature mentioned above, the sex differences in bullying has been well reported, and the associations between personality and bullying and between school climate and bullying have been also well explored, the relationship among the four psychological constructs has seldom and the mechanisms underlying the sex differences in bullying perpetration and victimization has not well established. Thus, the aim of the current study was to explore the sex differences in the perpetration and victimization of traditional school bullying among Chinese senior high school students and the mediating effects of Machiavellianism and the school climate, to further provide more constructive suggestions for prevention and intervention of school bullying.”

Method.

  1. More information about the sample is necessary, such as socio-economic information. 

Sampling information is missing. 

Response: Thanks for your suggestions and we accept the suggestion. We have added more information about the sample.

“2. Method

2.1. Participants and procedure

This study employed data from a high school in Zhengzhou, China. Participants of the survey were from 20 classes, which were randomly chosen from the school. After the written informed assent was obtained from the participants and the written informed consent was obtained from the students' guardians, the participants were assured that their answers were anonymous and confidential and were asked by teachers and well-trained psychological PhD students to completed the questionnaires. After removing invalid questionnaires, the effective sample size of this panel study was 727 (M = 16.8, SD = 0.9), consisting of 371 girls (48.3%) and 356 boys (46.4%). Senior high school Grade one students accounted for 53.8%, Grade two students 35.0%, Grade three students 11.2%. And 69.5% of the students have their household registrations in cities, 12.0% of the students in towns, 18.5% of the students in rural areas. The missing data of the non-demographic variables part was replaced by using the linear interpolation method. All eligible respondents were Han Chinese race. This project was reviewed and approved by the ethics committee at the authors' institution and had therefore been performed in accordance with the ethical standards laid down in the 1964 Declaration of Helsinki and its later amendments. The participants got paid after completed the questionnaires.”

  1. Dimensions of Machiavelism are in the method section but not defined in the introduction. 

Response: Thanks for your suggestions and we accept the suggestion. We have added the dimensions of Machiavelism in introduction (p2, line 69-71).

“According to the widely used Kiddie Mach scale (KMS) developed by Sutton and Keogh (2001), Machiavellianism has three dimensions: distrust of humanity, dishonest, and distrust.”

  1. Evidence about the validity of the scales is required. 

Response: Thanks for your suggestions and we accept the suggestion. We have added evidence about the validity of the scales, which is shown below in blue color.

“2.2. Measures

2.2.1. Chinese version of Kiddie Machiavellian Scale (KMS)

Pathological personality traits were measured using the brief version of KMS, which is composed of 16 items that assess distrust (five items; e.g., “Sometimes you have to hurt others in order to get what you want”), distrust of humanity (seven items; e.g., “Don’t tell anyone the real reason for doing something unless you have a special purpose”), dishonest (four items; e.g., “The best way to interact with someone is to say what they want to know”). Participants answered on a 4-point Likert scale (0 = “totally agree”, 3= “totally disagree”). Higher scores indicated lower levels of Machiavellianism. The scale has been proven to have good reliability and validity in Chinese samples (Yang et al., 2017). In this study, Cronbach's α coefficient is 0.71.

2.2.2. School Climate Perception Questionnaire

In the current study, the school climate perception was measured by the School Climate Perception Questionnaire (Jia et al., 2009), which contains 6 items (e.g., “There are students fighting in school”). Participants answered on a 4-point Likert scale (1= “never”, 4= “always”). The total score was sum of the scores of all items, with higher score indicating more negative perception of school climate. It has been confirmed to have good reliability and validity in Chinese samples (Wang et al., 2020). In this study, the Cronbach’s α coefficient is 0.73.

2.2.3. Chinese version of the Olweus Bully/Victim Questionnaire

The Chinese version of the Olweus Bully/Victim Questionnaire, mainly includes two dimensions: bullying perpetration (six items; e.g., “Give some classmates a bad name and scold them, or make fun of them”) and bullying victimization (six items; e.g., “Call me a bad nickname, scold me, or make fun of me”). Participants answered on a 5-point Likert scale (1= “Nothing happened in this semester”, 5 = “several times a week”). Higher scores indicated higher level of bullying perpetration or victimization. The Chinese version of the Olweus Bully/Victim Questionnaire has been recognized for its reliability and different forms of validity (Chang et al., 2020). In this study, the Cronbach’s α coefficient for bullying perpetration is 0.79, and bullying victimization 0.76.”

Results

1.Also is important to check this affirmation". Still, the direct effect of sex on bullying perpetration was not significant, which indicated that Machiavellianism played a mediating role in the association between sex and school climate" the relationship could be significant and play a mediating role. 

Response: Thanks for your kindly remainders and we are willing to accept the constructive advice. Please see the revised sentences in blue color below:

“but the direct effect of sex on bullying perpetration was not significant. The mediating effect was -0.113, with 95%CI [-0.214, -0.054] not including zero, which indicated that Machiavellianism played a mediating role between the association between sex and school climate.”

Discussion

1.In the results section, the authors indicate that the direct effect of sex on bullying perpetration was not significant. However, in the discussion, the authors affirm that "sex not only directly influences traditional school bullying perpetration and victimization behaviors but also indirectly via Machiavellianism and school climate." How is this possible? 

The authors should align the discussion to the results presented.

Response: Thanks for the careful review and posting the question. According to your comments, we checked our manuscript and made careful modifications in the discussion. Please see the revised sentences in blue color below:

“The present study explored differences in traditional school bullying perpetration and victimization behaviors between boys and girls and the effects of Machiavellianism and school climate. Consistent with our hypothesis, boys were more likely to be involved in the traditional school bullying perpetration and victimization, and sex indirectly influenced traditional school bullying perpetration and victimization behaviors via Machiavellianism and school climate. Furthermore, a novel chain-mediation model revealed that Machiavellianism and school climate mediated the effects of sex on traditional school bullying perpetration and victimization behaviors.”

  1. The authors must also explain the positive association between school climate and bullying and victimization in the discussion.

 Response: Thanks for your crucial suggestions and these contributed a lot to our draft. According to your comments, we have seriously re-written the discussion and explained the positive association between school climate and bullying victimization. Please check it in blue color like this:

“This study also reveals that as has hypothesized, the perception of school climate mediates sex differences in school bullying, which indicates that having a worse perception of school climate may also be the reason for the sex differences in bullying. The result was consistent with previous studies. For example, one study showed promoting a positive school climate could reduce bullying perpetration (Mucherah et al., 2018). Some researchers have found that a positive school climate protects adolescents against bullying victimization (Laftman et al., 2017; Wang et al., 2018). There is also research indicating that when students perceived a more positive school climate, the negative association between bullying victimization and student engagement was stronger(Yang et al., 2018). And students who attend schools with positive school climates are more likely to seek support upon experiencing victimization and threats of violence, compared to students with negative school climates (Eliot et al., 2010). In addition, some longitudinal studies have revealed that school climate, including student-teacher relationships, clear expectations, and fairness of rules, significantly predicts bullying victimization (Dorio, Clark, Demaray, & Doll, 2019; Wang et al., 2018). This result could be explained by the social ecological theory (Bronfenbrenner, 1979), which posts that bullying is not just the result of individual characteristics, but is influenced by multiple relationships with peers, families, and teachers, that is, school climate which has the most direct impact on development. Besides, individual biological differences, such as sex and race, shape interactions between the person and other individuals and opportunities present in the environment (Bronfenbrenner, 1989), which influences bullying perpetration and victimization (Bronfenbrenner, 1979).”

Submission Date

10 June 2022

Date of this review

28 Jun 2022 17:43:15

Reviewer 2 Report

11.  At line# 162, the score of Machiavellianism is in fact lower among boys than among girls; but meaning boys have higher Machiavellianism level than girls.

22.   At line# 272 - 273, authors need clarify how paying attention would reduce bullying behaviors among boys.

Author Response

Subject:

Sex differences in traditional school bullying perpetration and victimization among adolescents: A chain-mediating effect

Response letter

Dear editor,

Thank you for giving us the opportunity to revise our manuscript. We appreciate the reviewers for the helpful and constructive comments and suggestions, which are of great value to improving the quality of our article. Based on the comments, careful modifications have been made to our previous draft. In the current draft, all changes were highlighted within the document by blue colored text. After this revision, we also wrote a point-by-point response letter to the three reviewers’ comments.

We hope the revised manuscript will be acceptable for publication in the International Journal of Environmental Research and Public Health. Thank you very much for your time.

Reviewer 2

Open Review

(x) I would not like to sign my review report

( ) I would like to sign my review report

English language and style

( ) Extensive editing of English language and style required

( ) Moderate English changes required

(x) English language and style are fine/minor spell check required

( ) I don't feel qualified to judge about the English language and style

Yes

Can be improved

Must be improved

Not applicable

Does the introduction provide sufficient background and include all relevant references?

(x)

( )

( )

( )

Are all the cited references relevant to the research?

(x)

( )

( )

( )

Is the research design appropriate?

(x)

( )

( )

( )

Are the methods adequately described?

(x)

( )

( )

( )

Are the results clearly presented?

(x)

( )

( )

( )

Are the conclusions supported by the results?

(x)

( )

( )

( )

Comments and Suggestions for Authors

  1. At line# 162, the scoreof Machiavellianism is in fact lower among boys than among girls; but meaning boys have higher Machiavellianism level than girls.

Response: Thanks for posting the question, we checked the measurement for Machiavellianism, and found the sentence “the score of Machiavellianism is in fact lower among boys than among girls; but meaning boys have higher Machiavellianism level than girls.” was correct, because the lower score in indicated higher. Please check the descriptions of the Kiddie Machiavellian Scale (KMS). Additionally, to make the descriptions clearer, we changed the sentence a little. Please check the information below:

“2.2. Measures

2.2.1. Chinese version of Kiddie Machiavellian Scale (KMS)

Pathological personality traits were measured using the brief version of KMS, which is composed of 16 items that assess distrust (five items; e.g., “Sometimes you have to hurt others in order to get what you want”), distrust of humanity (seven items; e.g., “Don’t tell anyone the real reason for doing something unless you have a special purpose”), dishonest (four items; e.g., “The best way to interact with someone is to say what they want to know”). Participants answered on a 4-point Likert scale (0 = “totally agree”, 3= “totally disagree”). Higher scores indicated lower levels of Machiavellianism. The scale has been proven to have good reliability and validity in Chinese samples (Yang et al., 2017). In this study, Cronbach's α coefficient is 0.71.”

“Means, standard deviations among the primary variables in the study are shown in Table 1. Boys were found to score significantly lower on Machiavellianism (t = - 3.92, p <.001), which indicated that boys have higher Machiavellianism level than girls. Boys were found to score significantly higher on school climate (t = 4.03, p <.001), bullying perpetration (t = 2.58, p <.05), and bullying victimization (t = 4.78, p <.001), which indicated that boys have more negative perception of school climate, higher level of bullying perpetration and victimization than girls. ”

  1.  At line# 272 - 273, authors need clarify how paying attention would reduce bullying behaviors among boys.

 Response: Thanks for your kindly suggestions and we are willing to accept the constructive advices. According to your suggestions, we have made some modifications in the discussion and clarified how paying attention would reduce bullying behaviors among boys. Please see the revised sentences in blue color below:

“The present results revealed the mechanism underlying the sex differences in traditional school bullying perpetration and victimization among Chinese adolescents, and the findings have important implications for intervention for the school bullying. Firstly, boys are more likely to be involved in the traditional school bullying perpetration and victimization, the educators and parents could pay more attention to the school boys, rather than girls, when tried to reduce bullying phenomenon in the school. For example, parents and educators could conduct more bullying education than girls, give more encouragement to boys to deal with relationships with peers nonviolently and amicably and to further perceive the school climate more positively, and try to conduct more bullying prevention and intervention programs among boys.”

Submission Date

10 June 2022

Date of this review

21 Jun 2022 21:19:57

Reviewer 3 Report

The study is interesting and would have the potential for publication. However, there are some areas that require some improvements.

Firstly, I would recommend the authors have their papers edited carefully by a professional reviewer. Certain areas were not entirely clear. For example, what do you mean by "dark personalities"?

Secondly, the authors need to be more explicit about how their paper makes a unique contribution to the research literature. They should do a more critical review of the literature that discusses what's out there, what's missing and how this study would make a unique contribution.

Thirdly, their literature review needs to be germane to the research aims. Please check this and ensure you are reviewing literature relevant to your study aims rather than providing broad and general information.

Fourth, please provide information about the missing data and how they were handled.

Fifth, please describe the study procedure--how data were collected. Information about survey and how they were administered would be beneficial. 

Sixth, you say you propose three hypotheses, but I see four. Please double check this by having your paper edited carefully.

Seventh, please provide a rationale for your analyses being conducted.

Eighth, the discussion section is well written in general but please make sure you (a) discuss whether your results are consistent with prior study results, (b) discuss whether your results are consistent or inconsistent with your hypotheses, and (c) provide theoretical explanations for your results. 

Ninth, please do not end the paper with limitations. Please include implications for research and implications for clinical practice (or public health).

Author Response

Subject:

Sex differences in traditional school bullying perpetration and victimization among adolescents: A chain-mediating effect

Response letter

Dear editor,

Thank you for giving us the opportunity to revise our manuscript. We appreciate the reviewers for the helpful and constructive comments and suggestions, which are of great value to improving the quality of our article. Based on the comments, careful modifications have been made to our previous draft. In the current draft, all changes were highlighted within the document by blue colored text. After this revision, we also wrote a point-by-point response letter to the three reviewers’ comments.

We hope the revised manuscript will be acceptable for publication in the International Journal of Environmental Research and Public Health. Thank you very much for your time.

Reviewer 3

Open Review

(x) I would not like to sign my review report

( ) I would like to sign my review report

English language and style

( ) Extensive editing of English language and style required

(x) Moderate English changes required

( ) English language and style are fine/minor spell check required

( ) I don't feel qualified to judge about the English language and style

Yes

Can be improved

Must be improved

Not applicable

Does the introduction provide sufficient background and include all relevant references?

(x)

( )

( )

( )

Are all the cited references relevant to the research?

(x)

( )

( )

( )

Is the research design appropriate?

(x)

( )

( )

( )

Are the methods adequately described?

( )

(x)

( )

( )

Are the results clearly presented?

( )

(x)

( )

( )

Are the conclusions supported by the results?

(x)

( )

( )

( )

Comments and Suggestions for Authors

The study is interesting and would have the potential for publication. However, there are some areas that require some improvements.

Firstly, I would recommend the authors have their papers edited carefully by a professional reviewer. Certain areas were not entirely clear. For example, what do you mean by "dark personalities"?

Response: Thanks for the suggestions. We check the sentences in the manuscript carefully, and asked an expert in psychology to help improve the manuscript and changed some sentences that were not clear. As for the dark personalities, we made the following explanation and added it in the article.

“As one component of the Dark Triad, also dark personalities, which focuses on the socially aversive side of personality without being a clinical concept and is a composition of three conceptually distinct but empirically overlapping personality traits (Furnham, Richards, & Paulhus, 2013; Rahafar, Randler, Castellana, & Kausch, 2017), namely Machiavellianism, narcissism (refers to feelings of pride, superiority over others, admiration-seeking, egoism, and lack of empathy), and psychopathy (used to address mostly callousness, a lack of remorse, antisocial and impulsive behaviors (Rahafar et al., 2017)), Machiavellianism, characterized by a deceitful, materialistic, unemotional, and selfish stance (Christie, Geis, Festinger, & Schachter, 1970; Láng & Lénárd, 2015) is a behavioral tendency of individuals to use others to achieve personal goals (Zhao & Liao, 2013). According to the widely used Kiddie Mach scale (KMS) developed by Sutton and Keogh (2001), Machiavellianism has three dimensions: distrust of humanity, dishonest, and distrust.”

Secondly, the authors need to be more explicit about how their paper makes a unique contribution to the research literature. They should do a more critical review of the literature that discusses what's out there, what's missing and how this study would make a unique contribution.

Response: Thanks for your suggestions and we accept the suggestion. Please see the revised sentences in blue color are like this:

 “All in all, according the literature mentioned above, the sex differences in bullying has been well reported, and the associations between personality and bullying and between school climate and bullying have been also well explored, the relationship among the four psychological constructs has seldom and the mechanisms underlying the sex differences in bullying perpetration and victimization has not well established. Thus, the aim of the current study was to explore the sex differences in the perpetration and victimization of traditional school bullying among Chinese senior high school students and the mediating effects of Machiavellianism and the school climate, to further provide more constructive suggestions for prevention and intervention of school bullying.”

Thirdly, their literature review needs to be germane to the research aims. Please check this and ensure you are reviewing literature relevant to your study aims rather than providing broad and general information.

Response: Thanks for your suggestions and we are very happy to accept it. In the revised manuscript, we deleted some broad and general information, and added much information specific to the effects of dark personalities and school climate on bullying perpetration and victimization. Because there is too much information, please check the literature in the Introduction.

Fourth, please provide information about the missing data and how they were handled.

Response: Thanks for your crucial suggestions, which helped us a lot. According to your suggestions, we have added more information about the missing data. Please check the sentence shown below in blue color.

“The missing data of the non-demographic variables part was replaced by using the linear interpolation method.”

Fifth, please describe the study procedure--how data were collected. Information about survey and how they were administered would be beneficial. 

Response: Thanks for your kindly reminders. According to your comments, we have now added the study procedure.

“2. Method

2.1. Participants and procedure

This study employed data from a high school in Zhengzhou, China. Participants of the survey were from 20 classes, which were randomly chosen from the school. After the written informed assent was obtained from the participants and the written informed consent was obtained from the students' guardians, the participants were assured that their answers were anonymous and confidential and were asked by teachers and well-trained psychological PhD students to completed the questionnaires. After removing invalid questionnaires, the effective sample size of this panel study was 727 (M = 16.8, SD = 0.9), consisting of 371 girls (48.3%) and 356 boys (46.4%). Senior high school Grade one students accounted for 53.8%, Grade two students 35.0%, Grade three students 11.2%. And 69.5% of the students have their household registrations in cities, 12.0% of the students in towns, 18.5% of the students in rural areas. The missing data of the non-demographic variables part was replaced by using the linear interpolation method. All eligible respondents were Han Chinese race. This project was reviewed and approved by the ethics committee at the authors' institution and had therefore been performed in accordance with the ethical standards laid down in the 1964 Declaration of Helsinki and its later amendments. The participants got paid after completed the questionnaires.”

Sixth, you say you propose three hypotheses, but I see four. Please double check this by having your paper edited carefully.

Response: Thanks for your crucial suggestions. We have modified it into three hypotheses in the introduction. Please check them below in blue color:

“Based on the theoretical foundations and previous empirical researches, we proposed three hypotheses: (1) boys may be more likely to be involved in the traditional school bullying perpetration and victimization; (2) both of Machiavellianism and school climate would mediate the relationship between sex and traditional school bullying perpetration and victimization; (3) Machiavellianism and school climate may play a chain mediating effect on the relationship between sex and bullying perpetration and victimization (Fig. 1).”

Seventh, please provide a rationale for your analyses being conducted.

Response: Thanks for your crucial suggestions. In the current study, we conducted t test, correlation analysis, and mediating effects analysis. The reason for the t test was that the sex was a category variables, and t test could also help to show the relationships between sex and Machiavellianism, school climate, and bullying perpetration and victimization. As for the correlation analysis, the results could help to determine whether it was satisfied for further mediating effect analysis. In the mediating analyses which required the included variables being continuous variables, the four variables should be included in the model which had been hypothesized in the introduction, thus we transformed category variable, that is, sex, into a continuous variable. Besides, in order to get the standardized coefficients, all variables were standardized prior to the analysis.

In the revised manuscript, we deleted some sentences and changed some sentences, the revised main results were shown below:

“3. Results

3.1. Descriptive results

Means, standard deviations among the primary variables in the study are shown in Table 1. Boys were found to score significantly lower on Machiavellianism (t = −3.92, p <.001), which indicated that boys have higher Machiavellianism level than girls. Boys were found to score significantly higher on school climate (t = 4.03, p <.001), bullying perpetration (t = 2.58, p <.05), and bullying victimization (t = 4.78, p <.001), which indicated that boys have more negative perception of school climate, higher level of bullying perpetration and victimization than girls.

Table 1. Means, standard deviations, and sex differences among the primary variables.

Total

Boys

Girls

t

p

Variables

(n=727)

(n=356)

(n=371)

M±SD

M±SD

M±SD

Machiavellianism

43.03±6.00

42.20±6.17

43.93±5.73

-3.92

<.001

school climate

9.01±2.97

9.47±3.39

8.58±2.49

4.03

<.001

bullying perpetration

6.64±2.26

6.88±2.68

6.44±1.86

2.58

<.05

bullying victimization                            

7.37±2.82

7.90±3.44          

6.89±2.05            

4.78

<.001        

3.2. Correlation results

As shown in Table 2, Machiavellianism was significantly negatively correlated with school climate, traditional school bullying perpetration and victimization (r = −0.22, p < .001, r = −0.21, p < .001; r = −0.17, p < .001). However, school climate was significantly positively correlated with traditional school bullying perpetration and victimization (r = 0.25, p < .001; r = 0.34, p < .001), and bullying was significantly positively correlated with victimization (r = 0.53, p < .001).

Table 2. Pearson correlation coefficients between main variables (n=727).

Variables

1

2

3

4

1. Machiavellianism

1

2. school climate

-.21***

1

3. bullying perpetration

-21***

.25***

1

4. bullying victimization

-0.17***

.34***

.53***

1

Note: *p < .05, **p < .01, ***p < .001.

3.3. Chain-mediating effect

All variables were standardized prior to the analysis. Sex was dummy coded such that 0 = male and 1 = female. The chain mediation analyses were conducted using PROCESS for SPSS (model 6) (Hayes, 2013). ”

Eighth, the discussion section is well written in general but please make sure you (a) discuss whether your results are consistent with prior study results, (b) discuss whether your results are consistent or inconsistent with your hypotheses, and (c) provide theoretical explanations for your results. 

Response: Thanks for your crucial suggestions and these helped a lot to improve the quality of our draft. According to your suggestions, we have made some modifications in the discussion. Firstly, we have made modifications in the first paragraph of the discussion section (please check the changes which are shown in blue color), and now we make sure that our results presented are consistent with prior study results. Secondly, the results of our study are also consistent with our three hypotheses proposed in the introduction. Thirdly, we have provided four theoretical explanations for our results before: (a) the evolutionary approach and sexual selection theory were used to explain the sex differences in traditional school bullying perpetration and victimization; (b) the life history strategy (LHS) theory was used to explain that Machiavellianism mediated the sex differences in bullying; (c) the social ecological theory was used to explain the mediating role of school climate and chain mediating roles of Machiavellianism and school climate on the sex differences in the bullying perpetration and victimization. Please see them in the discussion part in blue color are like this:

“4. Discussion

The present study explored differences in traditional school bullying perpetration and victimization behaviors between boys and girls and the effects of Machiavellianism and school climate. Consistent with our hypothesis, boys were more likely to be involved in the traditional school bullying perpetration and victimization, and sex indirectly influenced traditional school bullying perpetration and victimization behaviors via Machiavellianism and school climate. Furthermore, a novel chain-mediation model revealed that Machiavellianism and school climate mediated the effects of sex on traditional school bullying perpetration and victimization behaviors.

The present study revealed the sex differences in traditional school bullying perpetration and victimization, and the sex had direct effect on bullying victimization, which is consistent with a previous study showing the level of traditional school bullying perpetration and victimization by boys were generally higher than girls (Pontes, Ayres, Lewandowski, & Pontes, 2018). Some researchers posted that aggressive tendency of male is the result of the interaction between biological and physiological factors, social and psychological factors and specific situations (Zhang, Gu, Wang, Wang, & Jones, 2000), and is also related to normative expectations and the anticipated consequences of aggression, that is, if boys display aggressive behaviors, they may suffer social consequences (Card, Stucky, Sawalani & Little, 2008). Besides, the evolutionary approach to bullying behavior treats bullying, like other common forms of aggressive behavior, as having costs and benefits and as being in some circumstances adaptive for an individual doing the bullying (e.g., by gaining resources or defending sub-group identity), even if not beneficial for the victim or the wider community. And the sex differences in bullying and aggression can be explained in terms of sexual selection theory, and effective strategies for damaging the status or reputation of males or females respectively (Pellegrini & Archer, 2005).

The present study showed that as had been hypothesized, Machiavellianism mediated the sex differences in bullying. Specifically, boys are prone to develop higher level of Machiavellianism with which one is more likely to possess self-focused goals and aggressive strategies (Yao & Enright, 2019), and then are more likely to be involved in the aggressive behaviors. This result also supported the life history strategy (LHS) theory, individuals higher in Machiavellianism, as the indicator of the "fast LHS" (Jonason & Tost, 2010; McDonald, Donnellan, & Navarrete, 2012), which is reflective of reproductive efforts over somatic efforts (Hurst & Kavanagh, 2017) tend to conduct aggressive and/or violent behaviors, as a measure to accomplish evolutionary domain-specific tasks, mainly mate attraction (Salas-Rodríguez, Gómez-Jacinto, & Hombrados-Mendieta, 2021). Researchers found that fast LHS is related to higher risk propensity in mate attraction, which in turn increases rule breaking, dangerous, destructive and illegal behaviors (Salas-Rodríguez et al., 2021).

This study also reveals that as has hypothesized, the perception of school climate mediates sex differences in school bullying, which indicates that having a worse perception of school climate may also be the reason for the sex differences in bullying. The result was consistent with previous studies. For example, one study showed promoting a positive school climate could reduce bullying perpetration (Mucherah et al., 2018). Some researchers have found that a positive school climate protects adolescents against bullying victimization (Laftman et al., 2017; Wang et al., 2018). There is also research indicating that when students perceived a more positive school climate, the negative association between bullying victimization and student engagement was stronger(Yang et al., 2018). And students who attend schools with positive school climates are more likely to seek support upon experiencing victimization and threats of violence, compared to students with negative school climates (Eliot et al., 2010). In addition, some longitudinal studies have revealed that school climate, including student-teacher relationships, clear expectations, and fairness of rules, significantly predicts bullying victimization (Dorio, Clark, Demaray, & Doll, 2019; Wang et al., 2018). This result could be explained by the social ecological theory (Bronfenbrenner, 1979), which posts that bullying is not just the result of individual characteristics, but is influenced by multiple relationships with peers, families, and teachers, that is, school climate which has the most direct impact on development. Besides, individual biological differences, such as sex and race, shape interactions between the person and other individuals and opportunities present in the environment (Bronfenbrenner, 1989), which influences bullying perpetration and victimization (Bronfenbrenner, 1979).

More importantly, the current study revealed as has been hypothesized, Machiavellianism and school climate played a chain-mediating effect on the sex differences in the bullying perpetration and victimization, which indicated that boys tend to display higher level of Machiavellianism, and then they are more likely to have negative perception of school climate, and ultimately involve in the bullying perpetration and victimization. The results were consistent with previous studies, for example, researchers have demonstrated psychopathy, which shared some common features with Machiavellianism, exerted an effect on the school climate relevant to bullying (Bandyopadhyay et al., 2009), which further affected bullying perpetration and victimization at school (Bayar & Ucanok, 2012; Gini, 2008; Kokkinos et al., 2010). The result supports the general aggression model posting individuals’ personality factors may affect their cognition and then affect their aggressive behaviors (Gomes & Eduardo, 2016). Past studies have showed that students are less likely to be engaged in aggression and bullying when they perceive their friends as trustworthy and helpful and their school climate as trusting and pleasant (Yang, Wang, & Lei, 2019; Wang, Iannotti, & Nansel, 2009). This result could also be explained by the social ecological theory (Bronfenbrenner, 1979), which posts that p shape interactions between the person and other individuals and opportunities present in the environment (Bronfenbrenner, 1989), which influences bullying perpetration and victimization (Bronfenbrenner, 1979).”

Ninth, please do not end the paper with limitations. Please include implications for research and implications for clinical practice (or public health).

Response: Thanks for your crucial suggestions, and we are very happy to accept them. According to your suggestions, we ended the paper with implications for research and implications for public health. Please check them in the discussion part in blue color are like this:

“6. Implications

The present results revealed the mechanism underlying the sex differences in traditional school bullying perpetration and victimization among Chinese adolescents, and the findings have important implications for intervention for the school bullying. Firstly, boys are more likely to be involved in the traditional school bullying perpetration and victimization, the educators and parents could pay more attention to the school boys, rather than girls, when tried to reduce bullying phenomenon in the school. For example, parents and educators could conduct more bullying education than girls, give more encouragement to boys to deal with relationships with peers nonviolently and amicably and to further perceive the school climate more positively, and try to conduct more bullying prevention and intervention programs among boys. Secondly, Machiavellianism and school climate may be the reasons for the sex differences in bullying perpetration and victimization, which indicated that intervention for reducing level of Machiavellianism and enhancing the positive level of school climate (for example, all school members are enlisted in an effort to combat bullying) may help reducing traditional school bullying perpetration and victimization among the adolescents. Thirdly, the current results showed that Machiavellianism and school climate played a chain mediating role in the relationship between the sex and bullying, which indicated that sex, Machiavellianism and school climate jointly affect the traditional school bullying perpetration and victimization among Chinese adolescents. Therefore, all these aspects should be taken into account in the future interventions to achieve the best intervention effect and finally to reduce the level of students’ bullying perpetration and victimization.

In conclusion, this study revealed mediating and chain-mediating effects of Machiavellianism and school climate on the sex differences in traditional school bullying perpetration and victimization among Chinese adolescents.”

Submission Date

10 June 2022

Date of this review

02 Jul 2022 05:36:06

Round 2

Reviewer 3 Report

The authors did a great job with the revisions and I believe it is publishable.